



# SegCloud: a novel cloud image segmentation model using deep Convolutional Neural Network for ground-based all-sky-view camera observation

Wanyi Xie[1, 2], Dong Liu[1, 2], Ming Yang[4], Shaoqing Chen[4], Benge Wang[4], Zhenzhu Wang[1, 2], Yingwei Xia[3], Yong Liu[2, 3], Yiren Wang[3] and Chaofang Zhang[3]

[1]Key Laboratory of Atmospheric Optics, Anhui Institute of Optics and Fine Mechanics, Chinese Academy of Sciences, Hefei 230088, China
[2]University of Science and Technology of China, Hefei 230026, China
[3]Opto-Electronics Applied Technology Research Centre, Institute of Applied Technology, Hefei Institutes of Physical Science, Chinese Academy of Sciences, Hefei 230031, China
[4]Anhui Air Traffic Management Bureau, Civil Aviation Administration of China, Hefei, 230094, China.

*Correspondence to*: Yiren Wang (wyiren90@mail.ustc.edu.cn) and Chaofang Zhang (zcf0413@mail.ustc.edu.cn)

**Abstract.** Cloud detection and cloud properties have significant applications in weather forecast, signal attenuation analysis, and other cloud-related fields. Cloud image segmentation is the fundamental and important step to derive cloud cover. However, traditional segmentation methods rely on low-level visual features of clouds, and often fail to achieve satisfactory performance. Deep Convolutional Neural Networks (CNNs) are able to extract high-level feature information of object and have become the dominant methods in many image segmentation fields. Inspired by that, a novel deep CNN model named SegCloud is proposed and applied to accurate cloud segmentation based on ground-based observation. Architecturally, SegCloud possesses symmetric encoder-decoder structure. The encoder network combines low-level cloud features to form high-level cloud feature maps with low resolution, and the decoder network restores the obtained high-level cloud feature maps to the same resolution of input images. The softmax classifier finally achieves pixel-wise classification and outputs segmentation results. SegCloud has powerful cloud discrimination ability and can automatically segment the whole sky images obtained by a ground-based all-sky-view camera. Furthermore, a new database, which includes 400 whole sky images and manual-marked labels, is built to train and test the SegCloud model. The performance of SegCloud is validated by extensive experiments, which show that SegCloud is effective and accurate for ground-based cloud segmentation and achieves better results than traditional methods. Moreover, the accuracy and practicability of SegCloud is further proved by applying it to cloud cover estimation.

## 1 Introduction

Clouds are among the most common and important meteorological phenomena, covering more than 66% of the global surface (Rossow and Schiffer, 1991; Carslaw and Ken, 2009; Stephens and Graeme, 2005; Zhao et al., 2019). The analysis of cloud condition and cloud cover plays a key role in various applications ( Zhang and Li, 2013; Zhang et al., 2013; Li et al., 2018; Ma et al., 2018; Bao et al., 2019; Li et al., 2015; Zhang et al., 2017). One can accurately acquire localized and simultaneous cloud condition because of the high temporal and spatial resolution of ground-based observed clouds. Many ground-based cloud measurement devices, such as radar and lidar, are used to detecting clouds (Zhao et al., 2014; Garrett and Zhao, 2013; Huang et al., 2012). Especially, ground-based all-sky-view imaging devices have been increasingly developed in decades (Long et al., 2001; Genkova et al., 2004; Feister and Shields, 2005; Tapakis and Charalambides, 2013) because of their larger



field of view and low cost. Accurate cloud segmentation is a primary precondition for cloud analysis of ground-based all-sky-view imaging equipment, which can improve the precision of derived cloud cover information and help meteorologists to further understand climatic conditions. Therefore, accurate cloud segmentation has become a hot research topic, and plenty of algorithms have been proposed for cloud analysis of ground-based all-sky-view imaging instrument in recent years (Long et

al., 2006; Kreuter et al., 2009; Heinle et al., 2010; Liu et al., 2014; Liu et al., 2015).

Traditional cloud segmentation methods usually use "color" as a measurement criterion to recognize clouds and clear sky. This is because that cloud particles have similar scattering intensity in blue and red bands, while the air molecules have more scattering intensity in blue band than that in red band due to the Mie scattering theory. So the blue and red channel values of a

cloud image is available as identifying features for cloud segmentation. For example, Long et al. (2006) and Kreuter et al. (2009) proposed a fixed-threshold algorithm that used the ratio of red and blue channel values to distinguish clouds and clear sky. In more detail, the pixels whose ratio of red and blue channel values are greater than the defined fixed threshold are identified as cloud, otherwise, the pixels are identified as clear sky. Similarly, Heinle et al. (2010) treat the difference of red and blue channel values as a judgement for cloud detection. Souzaecher et al. (2004) complement saturation as a characteristic

to identify the clouds on the basis of red and blue channel values. These fixed-threshold algorithms mentioned above, which strongly depend on the cameras' specifications and atmospheric conditions, are not adaptable for varied sky conditions (Long and Charles, 2010). Graph-cut method (Liu et al., 2015) and superpixel segmentation algorithm (Liu et al., 2014) are also applied for cloud segmentation to overcome the drawback of above fixed-threshold algorithms. Although certain improvement can be achieved, the performance of such algorithms still remains unsatisfactory in real measurement applications. Therefore,

accurate and robust cloud segmentation algorithms need to be well developed.

CNNs are outstanding and powerful object recognition technology, which have been applied to many fields widely in recent years, such as image and speech recognition (LeCun and Bengio, 1998; Taigman et al., 2014). CNNs also have achieved a breakthrough progress in cloud analysis too (Xiao et al., 2019; Shi et al., 2016; Liang et al., 2017; Shi et al., 2017) because of

its strong ability in cloud feature representation and advance cloud feature extraction to accurately identify clouds (Lecun et al., 2015). For example, Yuan et al. (2018) proposed an edge-aware CNN for satellite remote-sensing cloud image segmentation, which is proved to have superior detection results near cloud boundaries. Xiao et al. (2019) proposed an automatic classification model TL-ResNet152 to achieve the accurate recognition of ice crystal in clouds. Zhang et al. (2018) proposed a CloudNet model for ground-based observed cloud categorization, which can surpass the progress of the other

traditional approaches. However, few studies have evaluated the accuracy of CNNs in segmenting cloud images from ground-based all-sky-view imaging instrument. Almost no prior works have evaluated the reliability of the local cloud cover derived by the segmentation results of CNNs.



In this paper, we proposed a novel CNN model named SegCloud for accurately segmenting the cloud images from our self-made ground-based all-sky-view camera. The architecture of the proposed SegCloud is straightforward and clear, which possesses symmetric encoder-decoder structure followed by a softmax classifier. SegCloud can automatically segment the obtained whole sky images, and avoids the misrecognition caused by the traditionally color-based threshold methods because

of its powerful cloud discrimination. A new database is also created manually to train and test the SegCloud model, which consists of 400 whole sky images and the corresponding annotated labels. Extensive experimental results have shown that the proposed SegCloud model has effective and superior performance for cloud segmentation especially the advantage of recognizing the area near the sun. Moreover, the local cloud cover calculated by SegCloud model has high correlation with human observation, which not only further proves the accuracy of SegCloud but also provides a practical reference for future

automatic cloud cover observation.

The main contributions of this paper are the following:

- A novel CNN architecture is proposed for efficient and accurate whole sky image segmentation, outperforming the traditional cloud segmentation methods.

- A new cloud segmentation database is built for training the SegCloud model that includes 400 whole sky images and corresponding manual labels.

- The monthly cloud cover computed by SegCloud is compared with the human observation, and further prove the applicability of SegCloud in the meteorological practice.

The rest of this paper is organized as follows. In Sect. 2, the all-sky-view camera instrument and cloud segmentation database

we used in experiment are described. The SegCloud architecture is introduced in Sect. 3. The experimental details and results are presented in Sect. 4. Finally, conclusion and future work are depicted in Sect. 5.

## 2 Instrument and data description

In this work we use the sky images captured by the ground-based all-sky-view camera modeled as ASC100. The ASC100 is developed by Hefei Institute of Physical Science, Chinese Academy of Sciences (Tao et al., 2019), and is deployed on the

rooftop of Anhui Air Traffic Management Bureau, Civil Aviation Administration of China. The appearance and functional specifications of ASC100 are shown in Fig. 1. The basic imaging component of ASC100 is a digital camera equipped with a fish-eye lens with 180° field view angle. The digital camera faces the sky directly to capture whole sky images, and no traditional solar occulting devices, such as sun tracker or black shading strip, are required. To obtain whole sky images which have similar range of luminance to that observed through the human visual system, the digital camera snapshots and fuses ten

photos with different (from low to high) exposure time to one whole sky image using high dynamic range (HDR) technique. Fig. 1(b) presents the obtained whole sky image, which has RGB color with a resolution of $2000 \times 1944$ pixels. The ASC100 is programmed to automatically capture whole sky images every ten minutes during the daytime, and transfers those images

to a web-based data server in real time. The ASC100 device is completely sealed and the built-in fan and heater ensure the ASC100's internal environment stable.

To evaluate the cloud segment algorithm comprehensively, the whole sky images captured by ASC100 are collected, and a
new database called Whole Sky Image SEGmentation (WSISEG) database is built by ourselves, as shown in Figure 2(c). The WSISEG database consists of 400 whole sky images that covering various cloud cover under different weather condition. These images are resized to 480 × 450 resolution to accommodate the input size of CNN model, and their labels are manually created using photograph editing software. Different from the existing public database whose images are manually cropped from whole sky images and the areas near or including the sun and horizon are avoided (Li et al., 2011; Dev et al., 2017), our
database reflects complete whole sky condition. Thus the analysis of the whole sky condition can be achieved, and the performance of the segmentation algorithm in the areas near the sun and horizon also can be verified on our database. The annotated images of the proposed WSISEG database are divided into a training set (340 annotated images) and a test set (60 annotated images). The images from the test set are specially selected to cover different cloud types and cloud cover to comprehensively test the performance of the SegCloud model. The complex air condition images, such as cloud images under
heavy haze and fog, are not included in WSISEG database.

## 3 Cloud image segmentation approach

Detecting clouds from whole sky images remains challenging for traditional cloud segmentation methods based on low-level visual information. CNNs have the ability to mine the high-level cloud feature representation and are naturally considered as a novel choice for solving cloud segmentation problems. Thus, the SegCloud model is proposed and applied for whole sky
image segmentation. In this section, we first describe an overall layout of SegCloud model (Sect. 3.1), and then elaborate the training process of SegCloud model (Sect. 3.2).

### 3.1 SegCloud architecture

SegCloud is an optimized CNN model and mainly focuses on the end-to-end cloud segmentation task. As shown in Fig. 3,
SegCloud possesses symmetric encoder-decoder structure followed by a softmax classifier. SegCloud evolves from an improvement of VGG-16 network (Simonyan and Zisserman, 2014). The VGG-16 network is known as one of the best CNN architectures for image classification, which achieves huge success on ImageNet Challenge on 2014 and has been applied in many other fields (Wang et al., 2016). In this study, we improve the VGG-16 network and propose our SegCloud model, by replacing the fully connected layers of the original VGG-16 network with the decoder network, to achieve end-to-end cloud
image segmentation. SegCloud can take a batch of fixed-size whole sky images as inputs. The encoder network transforms the input images to high-level cloud feature representation, while the decoder network enlarges the cloud feature maps extracted



from the encoder network to the same resolution of input images. Finally the outputs of decoder network are fed to a softmax classifier to classify each pixel and produce the segmentation prediction.

### 3.1.1 Encoder Network

The encoder network of SegCloud consists of ten convolutional layers and five max-pooling layers. Each convolutional layer contains three operations - convolution, batch normalization (Ioffe and Szegedy, 2015), and Rectified Linear Units (ReLU) activation (Hinton, 2010). Firstly, the convolution operation accept the input feature maps and produces a set of output feature maps using a trainable filter bank with size of $3 \times 3$ window and a stride of 1. In order to accelerate the convergence of SegCloud model and alleviate the vanishing gradient problem during the later training process, the batch normalization are
then used to normalize these obtained feature maps. Then the following ReLU activation is applied to add the nonlinear expression ability of SegCloud model. In general, the shallow convolutional layers tend to capture the fine texture, such as shape and edge, while the deeper convolutional layers compute more high-level and complex semantic features using these obtained shallow layer features (Liang et al., 2017). The other essential content of encoder network is max-pooling layers. They are located separately after the convolutional layers, and achieve more translation invariance for robust cloud image
segmentation. Each max-pooling layer subsample the input feature maps with $2 \times 2$ window and a stride of 2. So the size of output feature maps is reduced by half and more predominant features are extracted. Through ten convolutional layers and five max-pooling layers, the high-level and small-size cloud feature maps are finally formed for further pixel-wise cloud segmentation.

### 3.1.2 Decoder Network

The goal of decoder network is to restore the obtained high-level cloud feature maps to the same resolution of input images and to achieve end-to-end cloud image segmentation. The decoder network consists of five up-sampling layers and ten convolutional layers. Each up-sampling layer up-samples their input feature maps and produces the sparse double-size feature maps, while each convolutional layer tends to densify these obtained sparse feature maps to produce more dense segmentation
results. In order to further ensure the segmentation accuracy of whole sky images, the up-sampling layer uses the feature information from the encoder network to perform up-sampling. As shown by black arrows in Fig. 3, the first four up-sampling layers use pooling indices from respectively corresponding max-pooling layers to perform up-sampling. The pooling indices is the locations of the maximum value in each max-pooling window of feature maps, which is proposed by Badrinarayanan et al. (2017) to make sure the accurate feature restoration with less computational cost during inference. Thus, we introduce
pooling indices into our first four up-sampling layers, and ensure effectively restore high-level cloud feature maps.

Although pooling indices can achieve less computational cost, they may lead to a slight loss of cloud boundary details. Both the high-level semantic information and the edge property play an important role to segment whole sky images. Thus, the last up-sampling layer directly use the whole feature maps duplicated by first max-pooling layers to perform up-sampling and



improve cloud boundary recognition. The specific operation are divided into three steps: (1) the up-sampling layer firstly uses bilinear interpolation method to up-sample the feature maps of nearest convolutional layers, and obtains output feature maps with the size doubled; (2) the feature maps from first max-pooling layer of encoder network are duplicated; (3) the feature maps obtained from (1) and (2) are concatenated to produce the final up-sampling feature maps. To sum up, through these five up-sampling layers, high-level cloud feature maps are restored to the same resolution of input images step by step.

### 3.1.3 Softmax classifier

The softmax classifier is located after decoder network to achieve final pixel-wise cloud classification, i.e. cloud image segmentation. Softmax classifier calculates the class probability of every pixel from feature maps, through the classification formula defined as follows:

$$\text{softmax}(x_i) = \frac{exp(x_i)}{\sum_j exp(x_j)} \qquad (1)$$

where $x_i$, $x_j$ is the feature value of the $i$th and $j$th class respectively. The output of the softmax classifier is a 3-channel probability image, where 3 is the number of categories (cloud, sky and background areas, such as sun and surrounding building). The predicted results of each pixel are the class with maximum probability. Thus, the final segment results output.

### 3.2 Training details of the SegCloud model

To apply SegCloud to cloud image segmentation, SegCloud must be trained firstly using the training set of WSISEG database. Before training the SegCloud, local contrast normalization is performed on the training set, to further accelerate training convergence of the SegCloud. Thereafter, SegCloud is trained with a mini-batch gradient descent using the TensorFlow software package named and running on an NVIDIA GeForce GTX1080. Each batch size is set as 10 and the training process uses momentum parameters with a decay of 0.9 (Sutskever et al., 2013). SegCloud is trained in 26,000 epochs with learning rate 0.006. SegCloud uses cross-entropy loss function defined in the following equation as the objective optimization function:

$$\text{loss} = -\frac{1}{N}\sum_{j=1}^{N} y_j^t \log y_j \,, \qquad (2)$$

where $N$ is the batch size, $y_j^t$ is the ground truth images, i.e. the corresponding labels from training set, and $y_j$ is the predicated segmentation of the input whole sky images. The computed cross-entropy loss values are continuously optimized using back propagation algorithm (Lecun et al., 2014) until the loss values converge. The final trained model and the best model parameters are saved for actual whole sky image segmentation.



## 4 Experiment

In order to evaluate the performance of the SegCloud model, we conduct extensive experiments on WSISEG database. Firstly, segmentation experiments are carried out to evaluate the effectiveness of the SegCloud. Furthermore, the superiority of the SegCloud is demonstrated by the comparison with the other traditional cloud segmentation methods. Finally, the accuracy and practicability of the SegCloud is further proved by applying the model to long-term observed cloud cover estimation.

### 4.1 Effectiveness of SegCloud model

To verify the effectiveness of the SegCloud in segmenting the whole sky images, a series test images are fed to the well-trained SegCloud model and the segmented images are obtained. Some representative segmentation results are illustrated in the second row of Fig. 4. The figure shows that the clouds, sky and sun, are marked in white, blue and black respectively, which denotes that the SegCloud successfully segments the whole sky images.

Furthermore, the effectiveness of the proposed SegCloud model is objectively quantified by calculating the segmentation accuracy. The labels of the test set are set to the ground truths and the segmentation accuracy of the SegCloud is calculated by comparing the segmentation result with the ground truth, as defined in Eq. (3). The average segmentation accuracy is also performed using Eq. (4).

$$Accuracy = \frac{T}{M} = \frac{T_{tcloud} + T_{tsky}}{M}, \tag{3}$$

$$Average\_accuracy = \frac{1}{n} \sum_{i=1}^{n} Accuracy_i , \tag{4}$$

where $T$ denotes the number of correctly classified pixels, which is the sum of true cloud pixels $T_{tcloud}$ and true clear sky pixels $T_{tsky}$; $M$ denotes the total number of pixels (excluding background regions) in the corresponding cloud image; and $n$ is the number of the test images. In this experiment, we have totally 60 test images, where 10 for clear sky images, 10 for overcast sky images and 40 for partial cloud images.

As reported in Table 1, SegCloud achieves a high average accuracy of 96.24%, which further objectively proves the effectiveness of the SegCloud model. Moreover, SegCloud performs very well on whole sky images with different cloud cover conditions which achieves 96.98% accuracy on clear sky images, 95.26% accuracy on partial cloud images, near-perfect accuracy 99.44% on overcast sky images. These experimental results show that the SegCloud is effective and accurate for cloud segmentation, which could provide a reference for future cloud segmentation research.

### 4.2 Comparison with other methods

To verify the superiority of the proposed SegCloud model, SegCloud is compared with two other conventional methods using the test set of the WSISEG database:





(1) R/B threshold method: considering the camera parameters and the atmospheric environment, the fixed ratio of red and blue channel values is set to 0.77 (Kreuter et al., 2009).

(2) Otsu algorithm(the adaptive threshold method): the threshold is automatically computed according to the whole sky images to be segmented. (Otsu, 2007).

We use the proposed SegCloud model, R/B threshold method and Otsu algorithm to segment the whole sky images of the test dataset. Some representative cloud segmentation results are presented in Fig. 4. It can be seen that Otsu algorithm has poorly performance to segment whole sky images, because it requires pixels of the same class to have similar gray value but clouds appear to be opposite. The R/B threshold method has better segmenting results compared with Otsu algorithm. However, the

segmenting accuracy is poor especially for circumsolar areas, as shown in Fig. 4(a), (b) and (c). The reasons may be that the circumsolar areas often have similar texture and intensity to the clouds due to the forward scattering of visible light by aerosols/haze and the dynamic range limitation of the detectors of the sky imager (Long and Charles, 2010). Traditional methods merely utilize the low-level vision cues to segment images, which tend to misclassify the pixels near the sun. However, SegCloud performs excellent segmentation than the other two conventional methods, especially the segmentation advantages

in the area near the sun. This is because that SegCloud learns from the given calibration database and constantly mines the deeper cloud features. So it has a better ability to identify circumsolar areas even though these pixels have textures and intensities that are similar to those of clouds.

To objectively verify the performance of the SegCloud, the average segmentation accuracy of three algorithms is also

calculated on the test dataset. As described in Table1, SegCloud obtains a much higher average accuracy than the other two methods, and achieves the best segmentation performance whatever for clear sky images, partial cloud images or overcast sky images, which shows the advantages of the proposed SegCloud model.

In summary, these experimental results show that the proposed SegCloud model outperforms conventional methods, and

provides more accurate cloud image segmentation.

### 4.3 Application of SegCloud in cloud cover estimation

To further verify the accuracy and practicability of SegCloud, we apply this model in cloud cover estimation and compare its derived cloud cover with human observation from 08:00 to 16:00 LT, July 1st to 31st in 2018. First, SegCloud derives the cloud cover information through segmenting corresponding whole sky images acquired by all-sky-view camera ASC100. For

the comparison purposes, the cloud cover of simultaneous human observation are provided by Anhui Air Traffic Management Bureau, Civil Aviation Administration of China, where ASC100 is located. The well-trained human observers archive cloud cover every hour both day and night through dividing the sky in oktas. Thus, the cloud cover estimated by SegCloud is multiplied by eight to be consistent with the human observation.

The total 279 pairs of cloud cover data are available and the comparison results are shown in Table 2. The correlation coefficient between SegCloud and human observation is high (0.84). Statistically, for all cases, the error (i.e. the cloud cover difference between SegCloud and human observation) within ±1/8 oktas is 75.3%, and the error within ±2/8 oktas is 90.9%. These results clearly show that the proposed SegCloud model has ability to accurately segment the whole sky images, and provides reliable cloud cover information. Moreover, the cloud cover derived by SegCloud and human observation have great agreement on the cases of clear sky (0/8 oktas), indicating SegCloud has better cloud segmentation performance for the circumsolar areas. These experimental results prove the accuracy and practicability of SegCloud and show its practical significance for the future automatic cloud cover observations.

## 5 Conclusion

In this study, a novel deep CNN model named SegCloud is proposed and applied to accurate cloud image segmentation. Extensive segmentation results demonstrate that the SegCloud model is effective and accurate. Due to its strong ability in cloud feature representation and advance cloud feature extraction, SegCloud outperforms the traditional methods significantly. Compared with human observation, SegCloud also provides reliable cloud cover information through computing the percentage of cloud pixel to all pixels, which further proves its accuracy and practicability. Moreover, we use photo editing software to construct a new database, which consisting of 400 whole sky images and identified labels. In our next work, the complex cloud images, such as images under heavy haze and fog, must be added to our database to improve its integrity. In addition, SegCloud will be further optimized using expanded database to improve the recognition accuracy in complex weather conditions. Based on cloud segmentation, besides the cloud cover, the cloud base altitude estimation and the cloud phase identification as well as the cloud movement are also crucial for meteorological applications and need to be elaborated in the future using our self-made all-sky-view camera.

Competing interests. No competing interests are present.

Data availability. Our database examples are available at https://github.com/CV-Application/WSISEG-Database

*Acknowledgments.* The authors appreciate the support provided by the Science and Technology Service Network Initiative of the Chinese Academy of Sciences (No. KFJ-STS-QYZD-022), the Youth Innovation Promotion Association CAS (No. 2017482) and the Research on Key Technology of Short-Term Forecasting of Photovoltaic Power Generation Based on All-sky Cloud Parameters (No. S201904b06020014).



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



Functional Specifications

| | |
|---|---|
| Field of view | 180˚ |
| Imaging period | 10 minutes |
| Image resolution | 2000×1944 |
| Operating temperature | -45˚ ~ 55˚ |
| Dimension and weight | 30×30×40 cm 30kg |
| Power supply | AC220V/50HZ |

(a)                    (b)                         (c)

Fig. 1: (a) the apperance of ASC100. (b) the whole sky images captured by ASC100. (c) the functional specifications of ASC100.

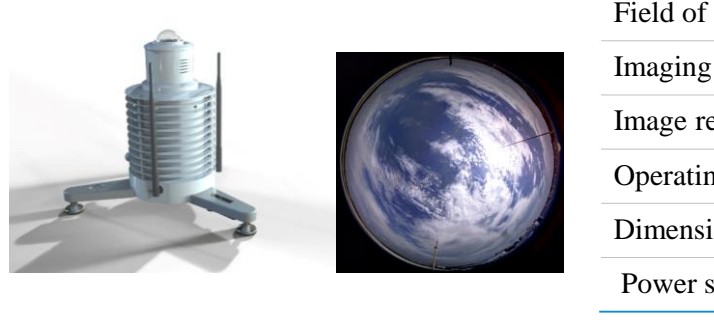

(a) SWIMSEG Database                    (b) HYTA Database

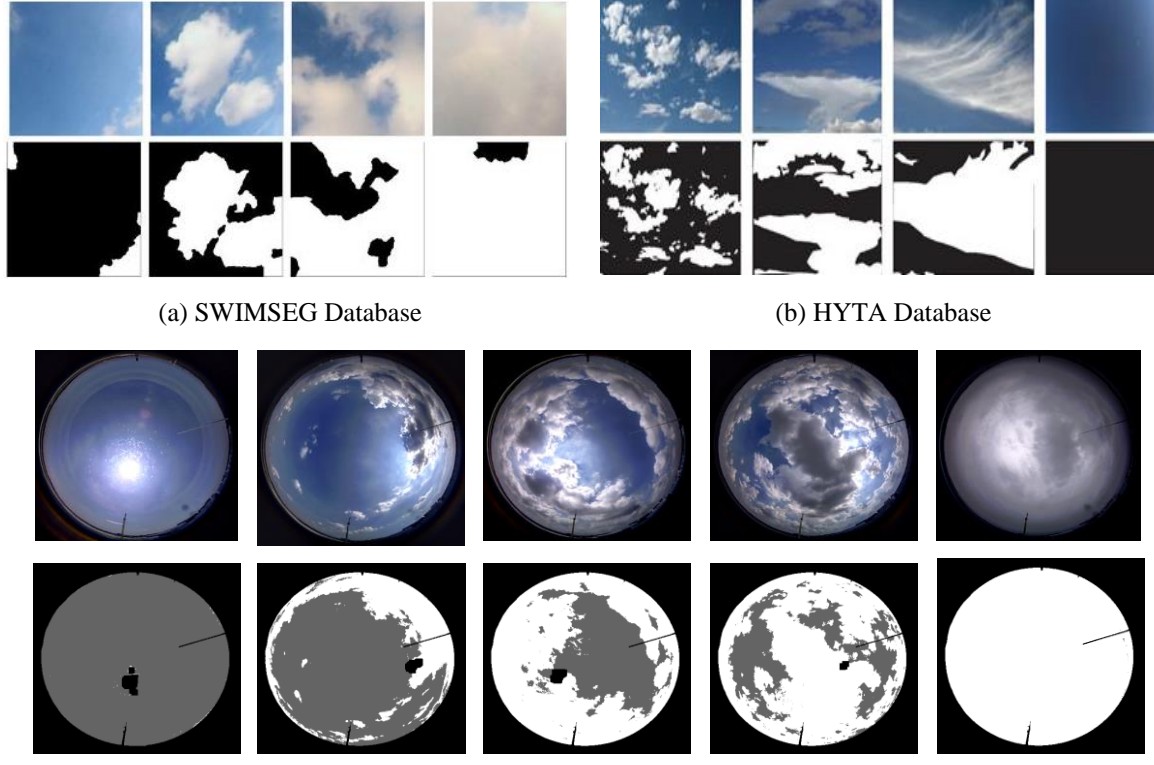

(c) WSISEG Database

**Figure 2: Representative sky images and their corresponding labels from (a). SWIMSEG database, (b). HYTA database and (c). WSISEG databases. The labels of the SWIMSEG and HYTA databases are binary images, where zero**





represents clear sky while one represents the cloud. While for the labels of the proposed WSISEG database, clouds, sky, and undefined areas (including sun and backgrounds) are marked with gray values 255, 100, and 0, respectively. Compared with SWIMSEG and HYTA database, the proposed WSISEG database has more advantages to reflect whole sky condition.

Figure 3: Illustration of the proposed SegCloud architecture. Overall, the networks contains an encoder network, a corresponding decoder network and a final softmax classifier.



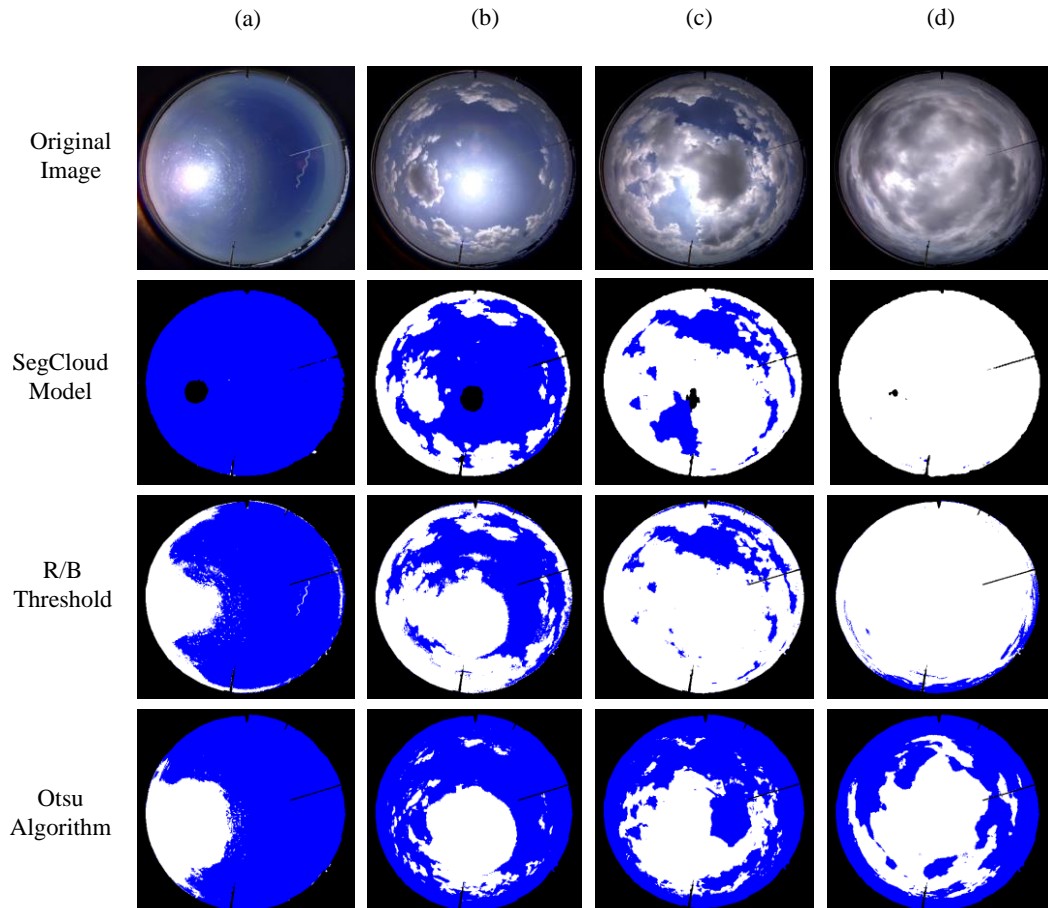

**Figure 4: Examples of segmentation results with three algorithms. The original images are presented in the top line, and the segmentation results of SegCloud model, R/B threshold approach, and Otsu algorithm are presented in the second, third, and last lines, respectively. Clouds, sky and sun are marked in white, blue and black. Masks are finally used in all result images to remove buildings around the circle for improved comparison of the sky and clouds.**

**Table 1: Segmentation accuracy (%) of three methods for clear sky images, partial cloud images and overcast sky images.**

| Method | Clear sky | Partial cloud | Overcast sky | Average |
|---|---|---|---|---|
| Otsu | 72.54 | 56.74 | 41.64 | 56.86 |
| R/B threshold | 70.63 | 81.36 | 90.95 | 81.17 |
| **SegCloud** | **96.98** | **95.26** | **99.44** | **96.24** |



**Table 2: Comparison of the derived cloud cover between SegCloud and human observation. The cloud cover provided by human observation are set to ground-truth, and the error is defined as the cloud cover estimated by SegCloud minus the ground-truth. The percentages of the error within ±1/8 oktas and the error within ±2/8 oktas are presented below. The correlation coefficient between SegCloud and human observation is also calculated.**

| Human observation (eighths) | Error within ±1/8 oktas (%) | Error within ±2/8 oktas (%) | Correlation |
|:---:|:---:|:---:|:---:|
| 0 | 97.0 | 100 | |
| 1 | 91.6 | 100 | |
| 2 | 64.7 | 100 | |
| 3 | 52.6 | 94.7 | |
| 4 | 50.0 | 75.0 | |
| 5 | 68.3 | 85.4 | |
| 6 | 83.3 | 91.7 | |
| 7 | 90.9 | 95.5 | |
| 8 | 98.1 | 100 | |
| **All** | **75.3** | **90.9** | **0.84** |

