# Peer review of "SegCloud: A novel cloud image segmentation model using deep convolutional neural network for ground-based all-sky-view camera observation"

_Atmospheric Measurement Techniques, 2019_

## Referee Comment (RC1) · Anonymous Referee #1 · 24 Dec 2019

Dear Editor: I am sending my comments with my colleague's one. please check them.

Best regards, Referee

This paper describes a proposal and results of an analysis system to retrieve cloud information from sky images taken by all-sky-view camera using a convolutional neural network(CNN). As well known, clouds have important roles for climate change as well as weather forecasting. Under these scientific interest, this study has been performed and is interesting for cloud researchers.

[Figure]

The paper summarizes a method of cloud segmentation from all-sky-view images and results using the newly developed method, compared with other traditional methods. As a result, the authors advocate that the method is more superior than the traditional ones, especially about solar aureole regions. In the usual photos including direct sun image, the light around these areas may be saturated as well as the sun itself. In this case, these have no information about cloud. Therefore, the cloud segmentation around these areas might be subject to the training data. As viewing the sample images, there are some doubtful regions found around the sun in the original sample image for "Clear sky" shown in Fig.4a (First row). While these look like small clouds or reflected images of dome, the system judges these are no cloud. On the other hand, an original image to be analyzed is fused with ten different exposure-photos (P3, L28-30). By using such a technique, it has a possibility to include cloud information around the sun. However, the paper shows no description on this effect when analyzing it. If the accuracy of cloud segmentation is improved by using the technique, please discuss it more, especially for aureole areas.

In the paper, authors have cited a paper written by Tao et al.(2019) introducing all-sky-view camera. In the Tao et al.'s paper they have discussed the analysis system of cloud segmentation called the optimized U-Net, which looks like similar to the present SegCloud. Because several authors are overlapped with those of that paper and one is the leading author of the present paper, they should discuss the difference of both systems and analyzed results in this paper clearly. Also Tao et al.(2019) have described the new database created for the system. This database must be the same as in this paper. Therefore, in this paper they should write "This database has been used in the present analysis, created in Tao et al. (2019)". The referee thinks that it is not suitable to include the contribution of the database production for training the SegCloud (P3, L15-16).

Tao et al.(2019) have discussed the relation between the U-Net results and the human observers' ones in detail. The similar discussion is found in the section 4.3 basically by
using the similar data frame (This paper used the data only of July, 2018, and Tao et al. used the data of August to November, 2018. The data site is the same.) If both analysis systems are different, it is useful and effective to discuss the difference between both results, but if not, this section may not be required in the paper because the detailed discussion has been already performed by using much more data.

I have asked my colleague with some comments of this paper, because he is an expert of this field. His comments are as follows,

The authors described a novel method for images segmentation based on CNN. They showed that the algorithm works well (i.e., good accuracy) if compared with the ground truth i.e., the images segmented by visual inspection. The CNN method provides improved results with respect to the traditional methods based on bands threshold. Further, they also give evidence of a good correlation with respect to cloud fraction computed by visual observation. Finally, the authors made public their WSISEG database. This is expected to be useful to the scientific community.

Major issues It is worth to mention that a paper describing the algorithm as well as some validation results have been recently published (Tao et al., 2019). I see that (a few) further details have been included in the present manuscript, but I don't believe that, at this stage, they can warrant a new publication. While the accuracy computed with respect to the ground truth is important and correctly reported, I would like to see more work regarding the comparison with other algorithms. The fixed threshold algorithm cannot be directly compared with the CNN output simply because it has been built to be used with a camera with a shadow band. Then, the actual fixed threshold algorithm (Long et al., 2006) is much more complicated than a simple R/B threshold shown in Fig. 4, and accounts (at least partially) for most of the issues mentioned in the current manuscript (e.g., solar obstruction). Your current comparison is clearly biased under clear sky and somewhat biased under partial cloud conditions. Then, the comparison performed under overcast sky makes sense, and indeed some studies used the R/B threshold method specifically to detect overcast sky occurrences (e.g., Dami-
ani et al., 2019). Therefore, I suggest comparing your results with additional/different more focused methods (e.g., ). Since Tao et al. (2019) already showed validations results with respect to visual observations for August-November 2018, please remove this part from the manuscript. I suggest including some applications exploiting the R/B method and its potentialities. For example, comparison with cloud fraction (or cloud mask) estimated by satellites (e.g., Himawari-8, MODIS. . .), analysis of trend/changes at different locations, adapting the same algorithm to images recorded by other cameras (since the method is based on CNN, it should be no so difficult).

Others The reference of the second segmentation method is Otsu (2007). However, I think that the authors actually refer to Otsu (1979) see https://ieeexplore.ieee.org/stamp/stamp.jsp?tp=&arnumber=4310076 Figure 2 -> The name of the two previous datasets showed in Fig. 2 are not mentioned in the text of the manuscript. In section 2, please include their names when their respective references are mentioned Figure 4 -> you should refer to the fixed R/B threshold algorithm with its appropriate reference (i.e. Long et al., 2006) in both Fig. 4 and main text Figure 4 -> please include also the original reference segmentation from the database

References mentioned above Long, C. N., Sabburg, J. M., CalboÌĄ, J., and PageÌĂs, D.: Retrieving cloud characteristics from ground-based daytime color all-sky images, Journal of Atmospheric and Oceanic Technology, 23(5), 633-652, 2006. Otsu, N.: A threshold selection method from gray-level histograms, IEEE Transactions on Systems, Man, and Cybernetics, 9(1), 62-66, 1979 Tao Fa, Wanyi Xie, Yiren Wang, and Yingwei Xia.: Development of an all-sky imaging system for cloud cover assessment, Applied Optics, 58, 5516-5524, 2019

Additional reference Damiani A., Hitoshi Irie, Tamio Takamura, Rei Kudo, Pradeep Khatri, Hironobu Iwabuchi, Ryosuke Masuda, Takashi Nagao, An intensive campaign-based intercomparison of cloud optical depth from ground and satellite instruments under overcast conditions, Scientific Online Letters on the Atmosphere, 15, 190-193,

DOI https://doi.org/10.2151/sola.2019-036, 2019

---

## Referee Comment (RC2) · Anonymous Referee #3 · 24 Dec 2019

This study proposed a novel deep CNN model named SegCloud and applied it to accurate cloud image segmentation. The segmentation results demonstrate that the SegCloud model is effective and accurate. It also demonstrated that SegCloud outperformed the traditional methods significantly. In principle, this is an interesting study with a new method useful for the science community. I would suggest its publication after necessary changes.

General comment: Significant efforts are needed for improvement of English writing.

[Figure]

Detail comments: Page 1: Line 29-31, one more reference could be added, Yang et al. (2017, doi:10.1002/2016JD025954), which shows the application of ground-based cloud observation for evaluating satellite-based observations. Line 32-34, "are used to detect clouds". Also, one more references could be added, Yang et al. (2018, doi: 10.1016/j.atmosres.2017.11.021). Page 2: Line 3-5, "... for recent years". Line 7-9, I am a little confused with this sentence. I understand that the aerosol particles along with some small cloud droplets follows Mie scattering. However, for air molecules, they generally follow Rayleigh scattering. Line 13, "treated" Line 14-15, this seems not a complete sentence. Line 21-23, "technology" -> "technologies". Also, this sentence seems with grammar error. Line 26-27, "is" -> "was" Page 3 Line 31, where is description for Fig. 1(a)? Page 4: Line 4-5, where is description for Fig. 2 (a)and (b)? Page 5 Line 7, "accepts" Line 9, "are" ->"is" Line 27-29, "are the locations", Line 30, "ensure effectively to restore ..." Line 32, "achieve ... cost ..."? Page 6 Line 13, "is"->"are" Line 14, the last sentence seems not a complete sentence. Line 17-20, please modify the description to make them more concise. Page 7 Line 7, "a series of ..." Lines 23-27, the performs are great. However, if you could provide some explanations or discussions regarding those that are not accurately classified, it would be more useful. Page 8 Line 7, why do you only provide "some representative segmentation results"? How do you make the choice of "some", subjectively or objectively? Line 7, "poorly" -> "poor" Line 13-14, do you mean "more excellent"/"more accurate"?
* * *

---

## Author Comment (AC2) · 22 Jan 2020

**Response Letter**

Dear referee:

Thank you for your comments and advices on our manuscript. We have carefully studied comments and made related revisions. Our response is as follows:

*Page 1:*
*1. Line 29-31, one more reference could be added, Yang et al. (2017, doi: 10.1002/2016JD025954), which shows the application of ground-based cloud observation for evaluating satellite-based observations.*

Reply: Thanks for your comment. The related reference has been added in Line 28.

*2. Line 32-34, "are used to detect clouds". Also, one more references could be added, Yang et al. (2018, doi:10.1016/j.atmosres.2017.11.021).*

Reply: Thanks for your comment. The sentence has been corrected to "are used to detect clouds", and the reference also has been added.

*Page 2*
*3. Line 3-5, "… for recent years".*

Reply: Thank you for pointing it out. The word "in recent years" has been corrected to "for recent years".

*4. Line 7-9, I am a little confused with this sentence. I understand that the aerosol particles along with some small cloud droplets follows Mie scattering. However, for air molecules, they generally follow Rayleigh scattering.*

Reply: Thank you for pointing it out. Now, the sentence has been corrected as "This is because that cloud particles have similar scattering intensity in blue and red bands because of the Mie scattering effort, while the air molecules have more scattering intensity in blue band than that in red band due to the Rayleigh scattering theory"

*5. Line 13, "treated" Line 14-15, this seems not a complete sentence.*

Reply: Thank you for your comment. The related content all have been changed to the past tense.

*6. Line 21-23, "technology" -> "technologies". Also, this sentence seems with grammar error.*

Reply: Thank you for pointing it out. The word "technology" has been corrected into

"technologies". The relative sentence has been corrected as "CNNs are outstanding and powerful object recognition technologies, which have been widely applied to many fields, such as computer vision and pattern recognition"

*7. Line 26-27, "is" -> "was"*

Reply: Thanks for your careful check. The word "is" has been corrected into "was".

*Page 3*
*8. Line 31, where is description for Fig. 1(a)?*

Reply: Thanks for your reminder. In Line 26, "The appearance and functional specifications of ASC are shown in Fig. 1", Fig. 1 actually means Fig. 1(a) and (c). Now, we has corrected the expression.

*Page 4:*
*9. Line 4-5, where is description for Fig. 2 (a) and (b)?*

Reply: Thanks for your reminder. We has added the description about Fig. 2(a) and (b) in Line 26-27.

*Page5*
*10. Line 7, "accepts"*

Reply: Thank you for pointing it out. The word "accept" has been corrected into "accepts".

*11. Line 9, "are" ->"is"*

Reply: Thank you for pointing it out. The word "are" has been corrected into "is".

*12. Line 27-29, "are the locations",*

Reply: Thank you for pointing it out. The word "is the location" has been corrected into "are the location".

*13. Line 30, "ensure effectively to restore ..."*

Reply: Thank you for pointing it out. The sentence has been corrected.

*14. Line 32, "achieve ... cost ..." ?*

Reply: Thanks for your comment. The sentence has been changed as "Although pooling indices have advantage in computational time, they may lead to a slight loss of cloud

boundary details".

*Page 6*
*15. Line 13, "is"->"are"*

Reply: Thank you for pointing it out. The word "is" has been corrected into "are".

*16. Line 14, the last sentence seems not a complete sentence.*

Reply: Thank you for pointing it out. The sentence has been corrected as "Thus, the final segment results are outputted".

*17. Line 17-20, please modify the description to make them more concise.*

Reply: Thanks for your suggestion. The related description has been modified as "Thereafter, SegCloud is trained on an NVIDIA GeForce GTX1080 hardware and the machine learning software package named TensorFlow. Mini-batch gradient descent is used as optimization algorithm to find the appropriate model weights. During the training process, the number of whole sky images fed to the SegCloud model per batch is 10 and momentum parameter with a decay of 0.9 is used (Sutskever et al., 2013). SegCloud is trained in 26,000 epochs with learning rate 0.006". We hope these expression would be more concise.

*Page 7*
*18. Lines 23-27, the performances are great. However, if you could provide some explanations or discussions regarding those that are not accurately classified, it would be more useful.*

Reply: Thanks for your suggestion. Although SegCloud model achieves great performance in whole sky image segmentation, it still has less recognition for very thin clouds. The related content has been added in Line 8-9, Page 8.

*Page 8*
*19. Line 7, why do you only provide "some representative segmentation results"? How do you make the choice of "some", subjectively or objectively?*

Reply: Thanks for your comment. In this work, we test 60 images, we cannot show all segmented results in the manuscript. So we randomly choose four whole sky images and their segmentation results under the precondition of including different cloud cover. Therefore, these four images separately show clear sky, partial cloudy sky and overcast sky.

*20. Line 7, "poorly" -> "poor"*

Reply: Thank you for pointing it out. The word "poorly" has been corrected into "poor".

*21. Line 13-14, do you mean "more excellent"/"more accurate"?*

Reply: Yes, the sentence has been corrected as "The R/B threshold method has more accurate segmenting results compared with Otsu algorithm".

---

## Author Response (AR1)

**Response Letter**

Dear editor and reviewers,

Sincerest thanks for your response and reviewers' comments on our manuscript entitled "SegCloud: A novel cloud image segmentation model using deep convolutional neural network for ground-based all-sky-view camera observation" (amt-2019-356). These comments are very valuable and helpful for revising and improving our paper, and they also have important guiding significance to our researches. We have studied comments carefully and made revisions which we hope meet with approval. This manuscript also has been modified by a professor whose native language is English to improve the English writing. Revised portion are marked in red in the resubmitted manuscript.

The main revisions in the paper and the responses to the reviewers' comments are as following:

Reviewer #1:

*1. As viewing the sample images, there are some doubtful regions found around the sun in the original sample image for "Clear sky" shown in Fig.4a (First row). While these look like small clouds or reflected images of dome, the system judges these are no cloud.*

Reply: Thanks for your concern. The sample image shown in Fig.4a is indeed the image of clear sky. The whole sky images used in experiment are captured by our all-sky imaging instrument as introduced in the Section 2. A plastic protective dome is added to the front of the imaging system to protect the device from dust and rain. However, after long-term field observation work, dust may fall on the protective dome and the protective dome has been some aged, which affects the imaging process and results in the existence of gray pixels that similar to cloud points.

*2. An original image to be analyzed is fused with ten different exposure-photos (P3, L28-30). By using such a technique, it has a possibility to include cloud information around the sun. However, the paper shows no description on this effect when analyzing it. If the accuracy of cloud segmentation is improved by using the technique, please discuss it more, especially for aureole areas.*

Reply: Thanks for your comment. The high dynamic range technique do guarantee better imaging and high-quality whole sky images, but it belongs to the part of the imaging system of all-sky imaging instrument. This manuscript presented in this paper mainly focuses on the cloud image segmentation algorithm, so the high dynamic range technique is not described in detail.

*3. In the Tao et al.'s paper they have discussed the analysis system of cloud*

*segmentation called the optimized U-Net, which looks like similar to the present SegCloud. Because several authors are overlapped with those of that paper and one is the leading author of the present paper, they should discuss the difference of both systems and analyzed results in this paper clearly.*

Reply: Thanks for your comment. The manuscript presented in this paper is different from the Tao et al.'s paper. Tao et al.'s paper focuses on the all-sky imaging instrument (ASC), and the ASC hardware system is mainly detailed, including the imaging system, data analysis module, et al. On the basis of these hardware system, the U-Net model is introduced into the cloud cover analysis. However, this paper focuses on the theoretical analysis of whole sky image segmentation algorithm. The new convolutional neural network model named SegCloud is mainly proposed for accurate whole sky image segmentation. The proposed SegCloud has been compared to other algorithms qualitatively and quantitatively according to ground truth images, which demonstrates its accuracy and superiority. Thus, these two papers are totally different.

*4. Tao et al. (2019) have described the new database created for the system. This database must be the same as in this paper. Therefore, in this paper they should write "This database has been used in the present analysis, created in Tao et al. (2019)". The referee thinks that it is not suitable to include the contribution of the database production for training the SegCloud (P3, L15-16).*

Reply: Thanks for your reminder. The data are actually mentioned and briefly described in Tao et al.'s paper, but no further details are presented. We realize it's not appropriate to emphasize the contribution in this paper. We have removed the relative content about the contribution of the database production in this paper, and have corrected the text to "In this paper, the database used in Tao et al. (2019) is applied to train and test the proposed SegCloud model". But in order to keep the integrity of the proposed algorithm, this manuscript still presents the database in detail, including its advantage and features.

*5. Tao et al. (2019) have discussed the relation between the U-Net results and the human observers' ones in detail. The similar discussion is found in the section 4.3 basically by using the similar data frame (This paper used the data only of July, 2018, and Tao et al. used the data of August to November, 2018. The data site is the same.) If both analysis systems are different, it is useful and effective to discuss the difference between both results, but if not, this section may not be required in the paper because the detailed discussion has been already performed by using much more data.*

Reply: Thanks for your comment. In Tao et al.'s paper, the ASC instrument has been running stably at the airport (data site) for cloud cover observation in real time. The accuracy of real-time cloud cover observation is affected by both the U-Net algorithm and the observation system. So, in Tao et al.'s paper, the purpose of the cloud cover comparison between the ASC instrument and human observer is to verify the stability and reliability of the ASC instrument. As to the manuscript presented in this paper, the

database used to train and test the proposed SegCloud algorithm is made by ourselves. To further demonstrate the persuasiveness and advantages of the proposed whole sky image segmentation algorithm, the human observer's cloud cover data is treated as ground-truth. At the same time, the whole sky images from ASC were downloaded and segmented, and then the cloud cover were computed and compared to with the data of human observation. As mentioned above, the purpose of the comparison with human observer's data in those two papers is different.

*6. While the accuracy computed with respect to the ground truth is important and correctly reported, I would like to see more work regarding the comparison with other algorithms. The fixed threshold algorithm cannot be directly compared with the CNN output simply because it has been built to be used with a camera with a shadow band. Then, the actual fixed threshold algorithm (Long et al., 2006) is much more complicated than a simple R/B threshold shown in Fig. 4, and accounts (at least partially) for most of the issues mentioned in the current manuscript (e.g., solar obstruction).*

Reply: Thanks for your comment. During the experiments, in addition to the R/B threshold algorithm and the Otsu algorithm, some other clustering algorithms were tested for whole sky image segmentation, such as k-means algorithm and mean-shift algorithm. None of those algorithms has satisfactory performance. The reasons for these results are that they require pixels of the same class to have similar gray value but clouds appear to be opposite (the same reason with Otsu algorithm, as introduced in Line 25, Page 7). We also test the other threshold segmenting algorithm using Red and Blue channel values, such as R-B, but their performance is similar to R/B threshold algorithm. So in this paper, we just pick two typical algorithms and compare our algorithm with them. For most cameras, in order to protect the CCD from the direct sunlight and avoid the large sun circle, shadow band will be added during the imaging process. Different from other cameras, our instrument uses the light-cutting module and high dynamic range technique, sun circle is small. So we think it's appropriate to use the R/B threshold algorithm as the comparison algorithm and it will not result in much bias. In Long et al.'s paper, the R/B threshold value is set as 0.6 through several tests performed on training images. But in our test experiment, we found the best threshold value is 0.77. Therefore, we choose 0.77 as the final threshold.

*7. Since Tao et al. (2019) already showed validations results with respect to visual observations for August-November 2018, please remove this part from the manuscript. I suggest including some applications exploiting the R/B method and its potentialities. For example, comparison with cloud fraction (or cloud mask) estimated by satellites (e.g., Himawari-8, MODIS), analysis of trend/changes at different locations, adapting the same algorithm to images recorded by other cameras (since the method is based on CNN, it should be no so difficult).*

Reply: Thanks for your suggestion. As we mentioned in comment #5, the purpose of comparison is different. Your suggestion all are great and they will be considered in our

further study. Thank you again.

*8. The reference of the second segmentation method is Otsu (2007). However, I think that the authors actually refer to Otsu (1979).*

Reply: Thanks for your reminder. We have corrected the reference information.

*9. The name of the two previous datasets showed in Fig. 2 are not mentioned in the text of the manuscript. In section 2, please include their names when their respective references are mentioned*

Reply: Thanks for your reminder. The names of the two databases have been added to the location where the respective references are mentioned.

Reviewer #3:

Page 1:
1. Line 29-31, one more reference could be added, Yang et al. (2017, doi: 10.1002/2016JD025954), which shows the application of ground-based cloud observation for evaluating satellite-based observations.

Reply: Thanks for your comment. The related reference has been added in Line 28.

2. Line 32-34, "are used to detect clouds". Also, one more references could be added, Yang et al. (2018, doi:10.1016/j.atmosres.2017.11.021).

Reply: Thanks for your comment. The sentence has been corrected to "are used to detect clouds", and the reference also has been added.

3. Line 3-5, "… for recent years".

Reply: Thank you for pointing it out. The word "in recent years" has been corrected to "for recent years".

4. Line 7-9, I am a little confused with this sentence. I understand that the aerosol particles along with some small cloud droplets follows Mie scattering. However, for air molecules, they generally follow Rayleigh scattering.

Reply: Thank you for pointing it out. Now, the sentence has been corrected as "Traditional segmentation methods generally use "color" as a distinguishing factor between clouds and clear sky because cloud particles have similar scattering intensity in blue and red bands due to the Mie scattering effort. By contrast, air molecules have

more scattering intensity in the blue band than in the red band due to the Rayleigh scattering theory".

5. Line 13, "treated" Line 14-15, this seems not a complete sentence.

Reply: Thank you for your comment. The related content all have been changed to the past tense.

6. Line 21-23, "technology" -> "technologies". Also, this sentence seems with grammar error.

Reply: Thank you for pointing it out. The word "technology" has been corrected into "technologies". The relative sentence has been corrected as "Convolutional neural networks (CNNs) are outstanding and powerful object recognition technologies, which have been widely applied in many fields, such as computer vision and pattern recognition"

7. Line 26-27, "is" -> "was"

Reply: Thanks for your careful check. The word "is" has been corrected into "was".

8. Line 31, where is description for Fig. 1(a)?

Reply: Thanks for your reminder. In Line 26, "The appearance and functional specifications of ASC are shown in Fig. 1", Fig. 1 actually means Figs. 1(a) and 1(c). Now, we has corrected the expression.

Page 4:
9. Line 4-5, where is description for Fig. 2 (a) and (b)?

Reply: Thanks for your reminder. We has added the description about Figs. 2(a) and 2(b) in Line 26-27.

Page5
10. Line 7, "accepts"

Reply: Thank you for pointing it out. The word "accept" has been corrected into "accepts".

11. Line 9, "are" ->"is"

Reply: Thank you for pointing it out. The word "are" has been corrected into "is".

12. Line 27-29, "are the locations",

Reply: Thank you for pointing it out. The word "is the location" has been corrected into "are the location".

13. Line 30, "ensure effectively to restore …"

Reply: Thank you for pointing it out. The sentence has been corrected.

14. Line 32, "achieve … cost …" ?

Reply: Thanks for your comment. The sentence has been changed as "Although pooling indices have advantage in computational time, they may lead to a slight loss of cloud boundary details".

15. Line 13, "is"->"are"

Reply: Thank you for pointing it out. The word "is" has been corrected into "are".

16. Line 14, the last sentence seems not a complete sentence.

Reply: Thank you for pointing it out. The sentence has been corrected as "Thus, the final segment results are outputted".

17. Line 17-20, please modify the description to make them more concise.

Reply: Thanks for your suggestion. The related description has been modified as "SegCloud is then trained on NVIDIA GeForce GTX1080 hardware and machine learning software package named TensorFlow. Mini-batch gradient descent is used as an optimization algorithm to find the appropriate model weights. During training, the number of whole-sky images fed to the SegCloud model per batch is 10, and momentum parameter with a decay of 0.9 is used (Sutskever et al., 2013)". We hope these expression would be more concise.

18. Lines 23-27, the performances are great. However, if you could provide some explanations or discussions regarding those that are not accurately classified, it would be more useful.

Reply: Thanks for your suggestion. Although SegCloud shows advantages in whole-sky image segmentation, some misidentification remains due to decreased recognition for extremely thin clouds, which should be investigated in the future. The related content has been added in Line 8-9, Page 8.

19. Line 7, why do you only provide "some representative segmentation results"? How do you make the choice of "some", subjectively or objectively?

Reply: Thanks for your comment. In this work, we test 60 images, we cannot show all segmented results in the manuscript. So we randomly choose four whole sky images and their segmentation results under the precondition of including different cloud cover. Therefore, these four images separately show clear sky, partial cloudy sky and overcast sky.

20. Line 7, "poorly" -> "poor"

Reply: Thank you for pointing it out. The word "poorly" has been corrected into "poor".

21. Line 13-14, do you mean "more excellent"/"more accurate"?

[revised manuscript text omitted]

---

## Referee Report (RR1)

*Review of amt-2019-356 "SegCloud: a novel cloud image segmentation model using deep Convolutional Neural Network for ground-based all-sky-view camera observation"*

I am glad that the author answered all my questions about this manuscript earnestly. I think the manuscript can be accepted after addressing the following comments.

Minor comments:

1. Page1 Line13: Delete "the" in front of the "weather forecast".

2. Page1 Line25: Change "is" to "are".

3. Page1 Line30: Change "condition" to "conditions".

4. Page1 Line31: Add "a" between "with" and "high".

5. Page1 Line33: Delete "the" in front of the "recent decades".

6. Page3 Line5: Add "a" between "has" and "high correlation".

7. Page3 Line26,28: Add "the" in front of the "WSISEG".

8. Page4 Line6: Change "process" to "processes".

9. Page5 Line9: Change "Upsampling" to "The upsampling". Add "the" between "ensure" and "segmentation".

10. Page5 Line14: Add "the" between "ensure" and "effective".

11. Page5 Line16: Add "an" in front of the "advantage".

12. Page5 Line18: Change "use" to "uses".

13. Page5 Line19: Add "the" in front of the "bilinear".

14. Page6 Line7,10,11: Add "the" in front of the "NVIDIA", "momentum", and "cross-entropy".

15. Page6 Line24: Change "is" to "are".

16. Page7 Line6: Change ";" to ",".

17. Page8 Line4: Change "objectively verify the performance of the SegCloud" to "verify the performance of the SegCloud objectively".

18. Page8 Line5: Delete "a" in front of the "higher".

19. Page8 Line10: Add "a" in front of the "human observation".

20. Page8 Line15: Add "the" in front of the "cloud cover".

21. Page8 Line21: Change "demonstrate" to "demonstrates".

22. Page8 Line29: Delete "the" in front of "SegCloud".

23. Page8 Line30: Change "pixel" to "pixels".

---

## Author Response (AR2)

Response Letter

Dear editor and reviewer,

Sincerest thanks for your response and reviewer's comments on our manuscript entitled "SegCloud: A novel cloud image segmentation model using deep convolutional neural network for ground-based all-sky-view camera observation" (amt-2019-356). The main revisions in the paper and the responses to the reviewer's comments are as following:

Reviewer #1:

1. *Page1 Line13: Delete "the" in front of the "weather forecast".*

Reply: Thank you for your correction. The word "the" has been deleted.

2. *Page1 Line25: Change "is" to "are".*

Reply: Thanks for your correction. The word "is" has been changed into "are".

3. *Page1 Line30: Change "condition" to "conditions".*

Reply: Thanks for your correction. The word "condition" has been changed into "conditions"

4. *Page1 Line31: Add "a" between "with" and "high".*

Reply: Thanks for your correction. The word "a" has been added.

5. *Page1 Line33: Delete "the" in front of the "recent decades".*

Reply: Thanks for your correction. The word "the" has been deleted.

6. *Page3 Line5: Add "a" between "has" and "high correlation".*

Reply: Thanks for your correction. The word "a" has been added.

7. *Page3 Line26, 28: Add "the" in front of the "WSISEG".*

Reply: Thanks for your correction. The word "the" has been added separately.

8. *Page4 Line6: Change "process" to "processes".*

Reply: Thanks for your correction. The "process" has been changed into "processes".

*9. Page5 Line9: Change "Upsampling" to "The upsampling". Add "the" between "ensure" and "segmentation".*

Reply: Thanks for your correction. The word "Upsampling" has been changed into "The upsampling", and the word "the" has been added between "ensure" and "segmentation".

*10. Page5 Line14: Add "the" between "ensure" and "effective".*

Reply: Thanks for your correction. The word "the" has been added.

*11. Page5 Line16: Add "an" in front of the "advantage".*

Reply: Thanks for your correction. The word "an" has been added.

*12. Page5 Line18: Change "use" to "uses".*

Reply: Thanks for your correction. The word "use" has been changed into "uses".

*13. Page5 Line19: Add "the" in front of the "bilinear".*

Reply: Thanks for your correction. The word "the" has been added.

*14. Page6 Line7, 10, 11: Add "the" in front of the "NVIDIA", "momentum", and "cross-entropy".*

Reply: Thanks for your correction. The word "the" has been added in front of the "NVIDIA", "momentum", and "cross-entropy" separately.

*15. Page6 Line24: Change "is" to "are".*

Reply: Thanks for your correction. The word "is" has been changed into "are".

*16. Page7 Line6: Change ";" to ",".*

Reply: Thanks for your correction. The punctuation ";" has been changed into ",".

*17. Page8 Line4: Change "objectively verify the performance of the SegCloud" to "verify the performance of the SegCloud objectively".*

Reply: Thanks for your correction. The sentence "objectively verify the performance of the SegCloud" has been changed into "verify the performance of the SegCloud objectively".

*18. Page8 Line5: Delete "a" in front of the "higher".*

Reply: Thanks for your correction. The word "a" has been deleted.

*19. Page8 Line10: Add "a" in front of the "human observation".*

Reply: Thanks for your correction. The word "a" has been added.

*20. Page8 Line15: Add "the" in front of the "cloud cover".*

Reply: Thanks for your correction. The word "the" has been added.

*21. Page8 Line21: Change "demonstrate" to "demonstrates".*

Reply: Thanks for your correction. The word "demonstrate" has been changed into "demonstrates".

*22. Page8 Line29: Delete "the" in front of "SegCloud".*

Reply: Thanks for your correction. The word "the" has been deleted.

*23. Page8 Line30: Change "pixel" to "pixels"*

Reply: Thanks for your correction. The word "pixel" has been changed into "pixels".

[revised manuscript text omitted]